# Review on Modeling and Control Strategies of DC–DC LLC Converters for Bidirectional Electric Vehicle Charger Applications

**Houssein Al Attar** [1,*] , **Mohamed Assaad Hamida** [1] , **Malek Ghanes** [1] **and Miassa Taleb** [2]

[1] Ecole Centrale de Nantes, LS2N, UMR CNRS 6004, 1 rue de la Noe, BP 92101, CEDEX 3, 44321 Nantes, France; mohamed.hamida@ec-nantes.fr (M.A.H.); malek.ghanes@ec-nantes.fr (M.G.)
[2] Renault Technocentre, 1 Avenue du Golf, 78280 Guyancourt, France; miassa.taleb@renault.com
* Correspondence: houssein.al-attar@ec-nantes.fr

**Abstract:** Bidirectional DC–DC converters are frequently chosen for applications requiring high power density such as in bidirectional electric vehicle (EV) chargers. Vehicle to Everything (V2X) technology makes the EV battery an electrical energy source. In this article, the use of a DC–DC LLC converter used in a bidirectional EV charger is reviewed. Different modeling approaches of the DC–DC LLC converter, such as small and large signal modeling, are discussed. Common modulation strategies applied to the DC–DC LLC converter in V2X mode, such as Pulse Frequency Modulation (PFM), Pulse Width Modulation (PWM) and Phase-Shift Modulation (PSM), are presented. The new challenge is to present the main characteristics and limitations of each modulation strategy in order to cover the whole operating zone of the EV charger in V2X mode. Furthermore, different control strategies based on a small or large signal model combined with different modulation strategies are highlighted. Linear and nonlinear controllers applied to the DC–DC LLC converter are discussed. Robust controllers are mainly highlighted regarding their advantage in ensuring the control robustness with respect to unexpected disturbances. A comparative study among modulation strategies as well as different control algorithms is conducted in terms of control performance and converter efficiency in V2X mode.

**Keywords:** electric vehicle charger; DC–DC LLC converter; V2X mode; small signal modeling; Pulse Frequency Modulation; Phase-Shift Modulation

## 1. Introduction

### 1.1. Overview

Understanding the advantages of electric mobility over the unsustainable usage of internal combustion engines (ICE) can clarify the reason why the popularity of electric vehicles (EVs) is increasing. The EV might have a huge impact on society. It represents the main support for e-mobility. Battery charging is an important element to take into account in the development of EVs. The onboard charger is generally embedded in the vehicle. The notion of Grid to Vehicle (G2V) describes how electric vehicle batteries are charged from the power grid. By integrating the bidirectional charger technology, it becomes possible to deliver energy in the other direction, from EV batteries to Grid (Vehicle to Grid or V2G) [1] or to Home/Load (V2H/L). The energy that has been stored in the battery can then be used as a source of current (in V2G) or voltage (in V2H/V2L). Vehicle to Everything (V2X) mode is the name given to the battery discharging operating mode. EVs, according to this concept, are not only electrical loads but also the electrical energy storage. Reactive and active power regulation, tracking of variable renewable energy sources, and load balancing are all features of an EV operating in V2X mode [2]. The battery charger must then be capable of ensuring the conversion of energy flow in both directions, becoming bidirectional and capable of both charging and discharging. As a result, EVs with bidirectional battery

chargers [3] can be connected into the smart grid, dramatically transforming the energy market. As shown in Figure 1, the EV charger can be considered as the association of two conversion stages [4]: an AC–DC part followed by an isolated DC–DC converter.

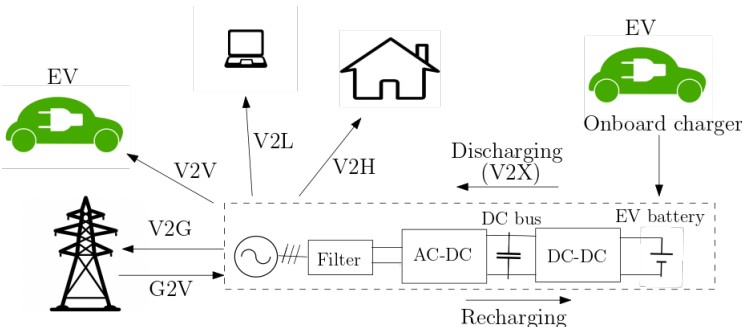

**Figure 1.** Bidirectional EV charger synoptic and operating modes.

Both functions (charging or G2V and discharging or V2X) are provided by the charger, presented in Figure 1, composed of bidirectional AC–DC and DC–DC converters. The AC–DC converter is a Power Factor Correction (PFC), which is used to ensure the grid current control. Meanwhile, the DC–DC converter is used to maintain the DC bus voltage and guarantee battery charging/discharging without any voltage ripples.

*1.2. Motivation*

The construction of interface systems, such as bidirectional chargers, between the grid and vehicle energy storage is one of the most technologically challenging aspects of implementing energy management systems. Battery charging is an important element to take into account for the development of EVs. The choice of the topology of power converters (AC–DC and DC–DC converters) defines the characteristics of the EV charger. In order to guarantee the bidirectional charger operation, the converters should be able to operate in the two directions with a high efficiency.

Bidirectional DC–DC converters are frequently chosen for applications requiring high power density. The bidirectional EV charger is one of the main applications of the bidirectional isolated DC–DC converters. In this technology, the DC–DC converters are presented between a DC bus link connected to an AC–DC converter and a high voltage battery pack as shown in Figure 1. The control strategy of the EV charger consists of two control algorithms: AC–DC and DC–DC controllers. The controllers should be implemented in order to ensure the grid, DC bus and battery stability.

Model-Based Design (MBD) approaches have been developed in trying to best meet charging requirements. MBD can result in considerable quality improvements in the design and control of power converters [5,6]. The aim of this review is to carry out an investigation on MBD and control strategies of the isolated DC–DC LLC converter implemented in a bidirectional EV charger toward improving the performance and efficiency.

*1.3. Related Works on Modeling and Control of Isolated DC–DC Converters*

In order to attain high efficiency throughout a broad operating range, Ref. [7] suggests a design optimization technique for a high-power off-board charger for EV applications. The electro-thermal modeling of the converter and control system design is the main emphasis of the design technique.

The design of a bidirectional onboard charger (OBC) for hybrid and electrified vehicles using a model-based approach is proposed in [8]. The design of a bidirectional single-phase OBC includes a DC–DC converter followed by an AC–DC converter. In order to assess the robustness of the implemented control algorithms, the produced models are tested under varied operating conditions.

Bidirectional DC–DC converters [9–11] are increasing in importance. Isolated DC–DC converters come in a variety of topologies and offer electrical isolation between the input and the output sides with a high-frequency transformer.

An interesting topology is Dual Active Bridge (DAB) converter [12,13]. Low device and component stresses, minimal switching losses (due to ZVS), high power density and efficiency, ad bidirectional power flow capabilities are all advantages of this topology.

Series resonant converters (SRCs) [14] also present different advantages. Because the resonant current and voltage are nearly sinusoidal due to the nature of resonance, those parameters can be approximated by merely its fundamental component without sacrificing too much precision for design purposes [14].

However, with the DAB and SRC topologies, soft-switching is impossible to maintain when the operating voltages at the ports are too high and under light load situations.

The isolated bidirectional DC–DC LLC resonant converter [15,16] is widely taken into account for control purposes in order to improve converter efficiency over a wide range of power and input/output voltages in industrial applications. On the other hand, because the resonant tank is not symmetric, there is a difference in the gain equation and the efficiency between the two operating modes (forward and reverse). Furthermore, while the DC–DC LLC converter is predicted to have a high efficiency in both operating modes, some of the unidirectional originator's intrinsic advantages are lost (reduced ZVS zone in the reverse mode resulting in lower efficiency), and large switching losses occur at high output voltages.

Establishing a representative dynamic model of the DC–DC LLC converter is necessary to create an appropriate control method.

To build small signal models in PWM converters, the state space average approach is commonly used in modeling methodologies. However, because the operating frequency is not constant and the energy is largely conveyed through the fundamental components of the voltage and current waveforms, it is not appropriate for a frequency-controlled resonant converter [17].

In [18,19], an MBD approach enables for the creation of a LLC transfer function, making control simple. However, it fails to account for nonlinear and uncertain effects, such as the DC–DC LLC converter structure, which impacts the robustness of the control law in the face of system perturbations and results in a reduced operating zone. The large signal model based on Extending Describing Function (EDF) [20–23] is proposed for the DC–DC LLC converter. When big signal transient disruption occurs, these models give sufficient dynamic information of the DC–DC LLC converter. However, because these models are complex and nonlinear, design of the control is more difficult. Small signal modeling [15,17,24] consists of using harmonic approximation to obtain more simplified model. Small signal transfer functions can be derived that are more in accordance with the design of control strategies. Many control laws, such as for PID controllers and sliding mode control, have been proposed in the literature for the DC–DC LLC converter based on large or small signal models. In [17], a simple PI controller is used to regulate the resonant current flowing in the LLC resonant circuit by varying the phase shift. In [25], a combined sliding mode–PI control strategy is implemented to reduce the chattering phenomenon. By adjusting the DC bus voltage, a PI controller is implemented to ensure battery voltage regulation in [26]. In [20,27], sliding mode control is used to insure the battery voltage stability by varying the respective switching frequency and duty cycle.

### 1.4. Contributions

Existing research works have studied control solutions for the DC–DC LLC converter related to the battery voltage or current control in forward mode (i.e., G2V mode). In most cases, the battery voltage varies over a reduced range and the converter operates at a narrow power zone. Furthermore, the existing control laws in the literature, combined with the different modulation strategies, are mostly based on the popular PI controller.

The main contribution is to provide a literature review of different modulation strategies applied to the DC–DC LLC converter in V2X mode in order to highlight the advantages and limitations of each and provide a comparative study in terms of converter efficiency, design and control complexity. Several modulation strategies, such as PFM, PWM and PSM, are presented.

On the other hand, the goal is to present modeling approaches such as large and small signal modeling applied to the DC–DC LLC converter in V2X mode. The benefits and the limitations of each strategy are highlighted. Based on such models, different control laws, which have been implemented, are discussed. Linear controllers based on a small signal model and nonlinear robust controllers based an a large signal model, combined with different modulation strategies, are presented. A comparative study between the different strategies is conducted with respect to the control performance and converter efficiency. The aim is to present existing control strategies in order to not only cover the whole operating zone in V2X mode but also to guarantee robustness against different disturbances and uncertainties in the DC–DC LLC converter system.

The paper is organized as follows: Section 2 provides some modeling approaches of the DC–DC LLC converter according to the modulation strategies (PFM, PWM and PSM). In Section 3, different control strategies based on a LLC model are presented. A general conclusion is drawn in Section 4.

## 2. Modulation System and Modeling Approaches in V2X Mode

### 2.1. System Presentation

Isolated bidirectional DC–DC converters [9–11] represent a crucial component for connecting storage devices between a DC bus and a battery with high-power uses. By discovering many topologies for the isolated bidirectional DC–DC converters, it was clear that each topology has the owner characteristics and the related limitations. Despite the simple design and implementation, the main drawback of the DAB topology is the limited voltage operation range [12]. Therefore, resonant topologies, such as the SRC and LLC topologies, are proposed to provide an improved soft-switching range.

The SRC topology [14] can improve the DAB operating zone, but it still suffers from soft-switching property loss at high input voltage and light load. With the CLLC converter [28], switches have the capability to operate with a wide ZVS zone in the two operating modes for a wide range of voltage gain. However, it has additional components, i.e., higher cost and sizing as well as more complex design and control. Thus, the trade-off between the control, design and efficiency requirements and the cost should be taken into account in order to define the converter topology. From here, DC–DC LLC resonant converters are frequently chosen [29–32] for uses that require a high power density, such as the EV charger.

The use of DC–DC LLC converters is common in numerous industrial applications [33]. The DC–DC LLC resonant converter is a popular isolated DC–DC topology because of its appealing advantages: Zero-Voltage Switching (ZVS) for power MOSFETs and Zero-Current Switching (ZCS) for diodes in a wide operating zone; simple circuit setup with low component count; and high power density with high switching frequency [34].

In this paper, the full-bridge DC–DC LLC resonant converter [16,35] and its control are investigated to suit the application features of the bidirectional EV charger, which include a wide range of battery voltages and high power densities [15,24].

The configuration of the full-bridge DC–DC LLC resonant converter is shown in Figure 2.

Figure 2 includes the DC bus capacitor, switching network, resonant circuit and battery model. Switches $S_1$–$S_8$ with anti-parallel diodes and snubber capacitors make up the switching network. A resonant inductor $L_r$, a magnetizing inductor $L_m$ and a resonant capacitor $C_r$ make up the resonant tank. The high-frequency transformer with turn ratios of $n$ ensures the galvanic isolation. The battery model is represented by the constant DC voltage source. It should be noted that there is an output filter circuit, which is not

represented in Figure 2, before the battery model that aims to reduce the battery current ripple. The voltage $V_{DC}$ is the DC bus voltage which represents the output voltage of the AC–DC stage of the EV charger, while $V_{bat}$ is the battery voltage and $P$ is the converter power.

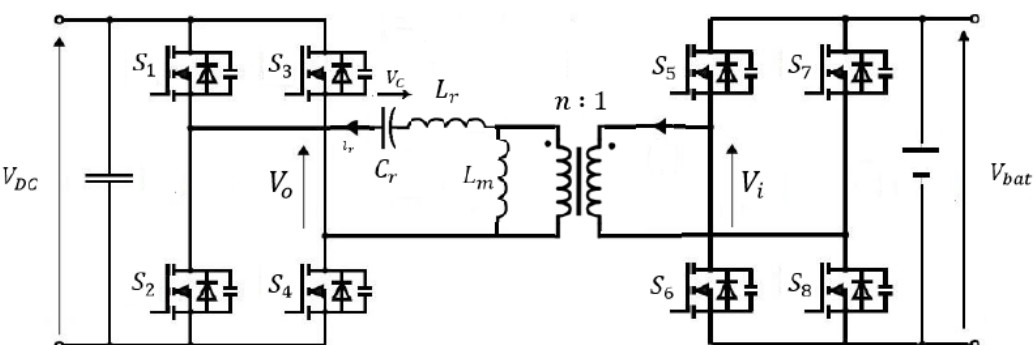

**Figure 2.** DC–DC LLC resonant converter.

Pulse Frequency Modulation (PFM) is the most widely used modulation technique for DC–DC LLC resonant converters. To ensure ZVS conditions, the switching frequency feasible zone is established between the minimum and highest authorized values. The EV charger's software and hardware implementation face a cost minimization challenge as a result of this frequency feasibility restriction. However, a wide operating switching frequency range is necessary to satisfy the system voltage gain requirement when the PFM strategy is used for wide input/output voltage and power ranges in the onboard battery charger in V2X mode. Due to the loss in the soft-switching operation brought on by the broad switching frequency range [32,36], the conversion efficiency and control performance are poor.

DC–DC LLC converters with the PFM strategy are not preferred in wide voltage range applications due to the following: large switching frequency variations are necessary that reduce the performance of the converter's Electro-Magnetic Interference (EMI); furthermore, in the case of low power loads, the PFM strategy causes degradation of efficiency and low control performance due to a wide switching frequency demand. As a result, DC–DC LLC converters that operate at a fixed switching frequency are favorable in applications involving a large voltage gain range and can effectively address the aforementioned problems. According to the existing literature, there are two categories of control techniques for DC–DC LLC converters that use fixed switching frequency: Phase-Shift Modulation (PSM) and Pulse Width Modulation (PWM). These two strategies have been investigated toward improving the control performance within a wide range of the battery voltage and high power density, as well as avoid the PFM strategy limitations (efficiency degradation at low power loads, switching frequency saturation, etc.).

LLC resonant converters contain the resonant circuit that consists of a series resonant inductor, capacitor and a parallel magnetic inductor. On the other hand, the LLC resonant converter is a nonlinear system due to the presence of switching frequency harmonics, nonlinear coupling between AC and DC model variables and the LLC resonant circuit. There are two modeling approaches: large signal modeling and small signal modeling. The first harmonic approximation (FHA) [36,37] is used to determine the voltage transfer functions. By disregarding the impact of the LLC resonant circuit dynamics, an averaged LLC converter model is demonstrated. Control can be made easy by using small signal modeling [18,19] to create an LLC transfer function based on an averaged mode [38]. It does not, however, take into consideration nonlinear and uncertain effects, such as the DC–DC LLC converter structure, which reduces the operating range of the control law and has an impact on the robustness of the control law in the face of system perturbations.

For the DC–DC LLC converter, models based on the Extending Describing Function (EDF) have been presented [20,21,23]. When an unexpected disturbance arises, these

models provide adequate LLC converter dynamic information. With these models, it is possible to create a mathematical model that describes the dynamics of the DC–DC LLC converter variables. The model is still nonlinear and complex, making control law design difficult.

### 2.2. Modulation System Based Small Signal Modeling

Small signal modeling with the first harmonic approximation (FHA) methodology [36] is used to examine the dynamics of the DC–DC LLC converter. The FHA is based on the following assumptions:

- The input voltage is modeled as an ideal sinusoidal voltage source, in which all higher-order harmonics are ignored and only the fundamental component is reflected;
- The capacitor of the output filter, the leakage inductance of the transformer and the effects of MOSFETs are ignored.

#### 2.2.1. PFM

The PFM strategy [39] consists of varying the switching frequency of MOSFET control signals where the full bridge's power MOSFETs are regulated in complementary mode at 50% duty. It should be noted that the PFM strategy is one of most frequently adopted modulation strategies for DC–DC resonant converters.

Assuming that the voltage signal can be only represented by the fundamental component in order to simplify the model design, the equivalent circuit model in G2V mode is presented in [40].

In V2X mode, while the full-bridge diodes on the primary side of the transformer are used for rectification, the power MOSFETs on the secondary side (S5-8) are regulated in complementary at 0.5 duty disregarding the dead time.

The input voltage of the LLC resonant circuit, in Figure 2, can be expressed in V2X mode, as in (1) [24]:

$$V_{if} = \frac{2\sqrt{2}}{\pi} n \, V_{bat} \tag{1}$$

The rectifier full-bridge side is driven by a square output voltage with a fundamental component $V_{of}$ expressed in (2):

$$V_{of} = \frac{2\sqrt{2}}{\pi} V_{DC} \tag{2}$$

Using small signal modeling with FHA, the equivalent model of the DC–DC LLC resonant converter (Figure 2) can be derived as presented in Figure 3 [24].

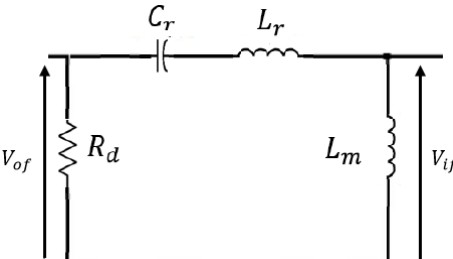

**Figure 3.** Equivalent model of the DC–DC LLC converter with FHA in V2X mode.

The equivalent resistor $R_d$ is defined, based on FHA [24], in (3):

$$R_d = \frac{8}{\pi^2} \frac{V_{DC}^2}{P} \tag{3}$$

Based on the equivalent model of the DC–DC LLC converter shown in Figure 3, the gain transfer function, based on FHA, can be expressed as in (4):

$$|G| = \frac{V_{of}}{V_{if}} = \frac{V_{DC}}{n\,V_{bat}} = \frac{|R_d.C_r\,wj|}{|1 - L_r\,C_r\,w^2 + R_d\,C_r\,wj|} \tag{4}$$

where $w = 2\pi f$ ($f$ is the switching frequency).

For the DC–DC LLC converter, the widely adopted modulation strategy is PFM. However, there is low conversion efficiency when the PFM strategy is adopted for wide voltage range application in the reverse operating mode. Therefore, many different modulation strategies such as PWM and PSM strategies are proposed to increase the DC–DC LLC converter feasibility operating with improved efficiency.

### 2.2.2. PWM

In the Pulse Width Modulation (PWM) approach [41], a rectangular pulse wave with variable pulse width is employed. The key idea behind the PWM technique is that the duty cycle can change the square wave input voltage of the resonant circuit.

The fundamental idea behind the PWM technique is that the square input voltage of the LLC resonant circuit can be adjusted by modifying the duty cycle while maintaining a constant switching frequency. The gain value is affected by the resonant tank impedance (Equation (4)), which depends on the switching frequency. The PWM strategy is designed based on FHA. The switching frequency $f$ is constant and the variable duty cycle $D$ ensures DC bus voltage control.

PWM offers the benefit of being able to work at switching frequencies that are lower than those of the traditional PFM control method.

The following expression, which relies on $D$, describes the fundamental component of the resonant circuit input voltage [27]:

$$V_{if} = \frac{\sqrt{2}}{\pi}\,n\,V_{bat}\,[1 - cos(2D\pi)] \tag{5}$$

The following is the fundamental component of the square output voltage:

$$V_{of} = \frac{2\sqrt{2}}{\pi}\,V_{DC} \tag{6}$$

Based on FHA, the transmission gain of the resonant tank for the PWM mode $G'$ [27] is provided by:

$$|G'| = \frac{V_{DC}}{n\,V_{bat}} = \frac{1 - cos(2\pi D)}{2}\,|G| \tag{7}$$

where $G$ is defined in Equation (4).

On the other hand, in order to respect FHA and ensure appropriate resonant operation of the DC–DC LLC converter, the duty cycle variation should be limited. In cases of high duty cycles, the ZVS property can be lost.

### 2.2.3. PSM

The Phase-Shift Modulation (PSM) technique [42] is a method of modifying the phase-shift angle between two signals with a constant signal frequency. The converter is controlled by varying the phase shift between the MOSFET control signals, with fixed switching frequency and fixed duty cycle.

The PSM strategy is a very effective technique for the DC–DC LLC converter, overcoming the drawbacks of the PFM strategy and avoiding the limitations of the PWM strategy.

By adjusting the phase shift between the control signals of the MOSFETs, which have constant switching frequency and duty cycle, the DC–DC LLC gain can be changed. The input voltage's fundamental harmonic is impacted by the controlled phase shift.

The duty cycle is set to 0.5 for all MOSFETs, where the controlled phase-shift angle $\theta$ is specified, as shown in Figure 4, as the phase-shift angle between the left (MOSFET S6 (inversely S5)) and right arms (MOSFET S7 (inversely S8)) in the identical H-bridge in Figure 2.

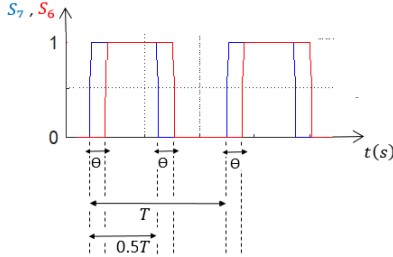

**Figure 4.** MOSFET signals in the case of the PSM strategy.

The fundamental component of the input voltage of the LLC resonant tank has the following expression, which relies on $\theta$, based on small signal modeling (FHA) and using asynchronous clamped mode (ACM) [31,43]:

$$V_{if} = \frac{n\,V_{bat}}{\sqrt{2}\pi} \sqrt{[10 + 6cos(\theta\pi)]} \tag{8}$$

The fundamental component of the LLC resonant tank output voltage is:

$$V_{of} = \frac{2\sqrt{2}}{\pi} V_{DC} \tag{9}$$

The LLC transmission gain of the resonant circuit $G''$ in case of the PSM strategy is then given by:

$$|G''| = \frac{V_{DC}}{n\,V_{bat}} = \frac{\sqrt{(10 + 6cos(\theta\pi)}}{4} |G| \tag{10}$$

where $G$ is expressed in Equation (4) and the phase shift $\theta$ is adjusted between 0 and 0.5.

To summarize, the main characteristics and limitations of different modulation strategies, applied to the DC–DC LLC converter for an EV charger application in V2X mode, are shown in Table 1.

**Table 1.** Comparison table of modulation strategies.

|                    | **Characteristics** | **Limitations** |
| --- | --- | --- |
| PFM [24,39,44,45]  | Variable switching frequency<br>Fixed duty cycle at 0.5<br>ZVS inside the non-saturated zone<br>High efficiency at high power loads | Important frequency saturation zone<br>Low efficiency at low power loads<br>Important tracking error in saturation zone |
| PWM [27,41]        | Variable duty cycle<br>Fixed switching frequency<br>Cover more operating points than PFM<br>High efficiency at high power loads | No ZVS with duty cycles far from 0.5<br>Low efficiency at low power loads |
| PSM [31,42,46,47]  | Variable phase-shift angle<br>Fixed switching frequency<br>Fixed duty cycle at 0.5<br>Cover the whole operating zone<br>Improved efficiency at low power loads<br>High efficiency at high power loads | Higher overshoot percent<br>Requires soft-start strategy |

The FHA model based PFM strategy represented by Equation (4) is used to obtain the controlled switching frequency as detailed in [15,24]. In order to ensure the ZVS

operation, the switching frequency should be limited. For a wide operating zone in V2X mode, especially under light load conditions (for low power loads), the PFM strategy is not sufficient for reaching all the operating points because the switching frequency is saturated, causing low conversion efficiency. The PWM strategy, represented in Equation (7), is able to reach more operating points by fixing the switching frequency and varying the duty cycle. However, a wide operating zone requires a large duty cycle variation that is not coherent with the FHA principle and causes reduced efficiency. Then, the model based on the PSM strategy (10) is proposed to avoid the limitations of PFM and PWM by fixing both switching frequency and duty cycle and varying the phase-shift angle in order to reach the whole operating zone in V2X mode with improved efficiency.

*2.3. Large Signal Modeling*

2.3.1. PFM

The large signal model can be split into three parts [48]: write the nonlinear dynamic state equations, use variables' harmonic approximation and apply the Extended Describing Function (EDF) [20–23] for approximation of nonlinear terms.

The equivalent circuit of the DC–DC LLC converter in V2X mode is presented in Figure 5.

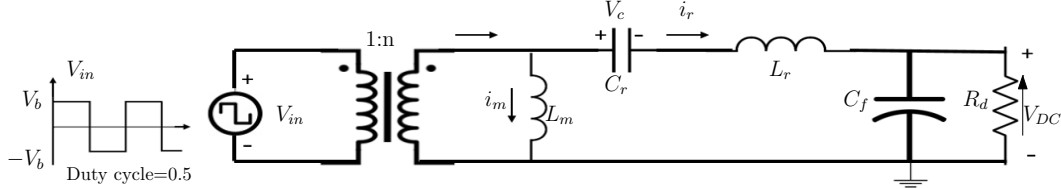

**Figure 5.** LLC equivalent circuit in V2X mode.

The LLC converter resonant tank receives a square wave voltage created by the full bridge. The output capacitor's parasitic resistor is ignored [49].

The nonlinear dynamic equations, based on the LLC equivalent circuit in Figure 5, can be defined [20,23,49] by applying the Kirchhoff's laws, as in Equations (11)–(13):

$$\frac{di_r}{dt} = \frac{1}{L_r}V_{in} - \frac{1}{L_r}V_c - \frac{1}{L_r}sign(i_r)V_{DC} \tag{11}$$

$$\frac{dV_c}{dt} = \frac{1}{C_r}i_r \tag{12}$$

$$C_f\frac{dV_{DC}}{dt} = |i_r| - \frac{1}{R_d}V_{DC} \tag{13}$$

where $V_{in}$ is a square wave voltage produced by the full-bridge switches and applied to the resonant tank in V2X mode. The resonant current $i_r$, capacitor voltage $V_c$ and DC bus voltage $V_{DC}$ are also state variables.

The AC state variables are decomposed into sine and cosine components, and the derivatives are equal to zero to yield the steady-state values using the sinusoidal approximation. On the other hand, this decomposition provides two states for each AC variable, resulting in a higher-order dynamic model. The approximation for the series resonant current and its derivative can be presented as follows in Equations (14) and (15), respectively [20,23,49].

$$i_r(t) = i_{rs}(t)\sin(w_s t) - i_{rc}(t)\cos(w_s t) \tag{14}$$

$$\frac{di_r}{dt} = \left(\frac{di_{rs}}{dt} + w_s i_{rc}\right)\sin(w_s t) - \left(\frac{di_{rc}}{dt} - w_s i_{rs}\right)\cos(w_s t) \tag{15}$$

Likewise, $V_c$ and $\frac{dV_c}{dt}$ can be divided into sine and cosine components, as shown in Equations (16) and (17):

$$V_c(t) = V_{cs}(t)\sin(w_s t) - V_{cc}(t)\cos(w_s t) \tag{16}$$

$$\frac{dV_c}{dt} = \left(\frac{dV_{cs}}{dt} + w_s V_{cc}\right)\sin(w_s t) - \left(\frac{dV_{cc}}{dt} - w_s V_{cs}\right)\cos(w_s t) \tag{17}$$

It is worth noting that nonlinear terms such as $sign(i_r)$ and $|i_r|$ appear in the dynamic Equations (11)–(13). The EDF concept is a strong mathematical tool for modeling and studying the dynamic behavior of resonant converters. By breaking modulated waveforms into sine and cosine waveforms, this method combines time domain and frequency domain analysis to derive the model. The fundamental harmonic terms can be used to approximate the nonlinear terms.

The nonlinear terms could then be expressed by their sine and cosine components using EDF [20,22,23,49,49] as presented in Equations (18)–(20):

$$V_{in}(t) = F_1(d, V_{in})\sin(w_s t) \tag{18}$$

$$sign(i_r) = F_2(i_{rs}, i_p)\sin(w_s t) - F_3(i_{rc}, i_p)\cos(w_s t) \tag{19}$$

$$|i_r| = F_4(i_{rs}, i_{rc}) \tag{20}$$

where the EDF parameters are F1 $(d, V_{in})$, F2 $(i_{rs}, i_p)$, F3 $(i_{rs}, i_p)$ and F4 $(i_{rs}, i_{rc})$. The variable $i_p$ is defined as in Equation (21):

$$i_p = \sqrt{i_{rs}^2 + i_{rc}^2} \tag{21}$$

It should be noted that $d$ is the duty cycle, which is fixed at 50%.
The EDF parameters are defined as in Equations (22)–(25).

$$F_1(d, V_{in}) = \frac{2\sqrt{2}}{\pi} n V_{bat} \sin \pi d = \frac{2\sqrt{2}n V_{bat}}{\pi} \tag{22}$$

$$F_2(i_{rs}, i_p) = \frac{4}{\pi}\frac{i_{rs}}{i_p} \tag{23}$$

$$F_3(i_{rc}, i_p) = \frac{4}{\pi}\frac{i_{rc}}{i_p} \tag{24}$$

$$F_4(i_{rs}, i_{rc}) = \frac{2}{\pi}i_p \tag{25}$$

By making use of EDF terms and the harmonic approximations, splitting the sine and cosine terms, the following Equations (26)–(30) are obtained [20,23,49]:

$$\frac{di_{rs}}{dt} = -w_s i_{rc} - \frac{V_{cs}}{L_r} - \frac{4 i_{rs} V_{DC}}{\pi L_r i_p} + \frac{2\sqrt{2}n V_{bat}}{\pi} \tag{26}$$

$$\frac{di_{rc}}{dt} = w_s i_{rs} - \frac{V_{cc}}{L_r} - \frac{4 i_{rc} V_{DC}}{\pi L_r i_p} \tag{27}$$

$$\frac{dV_{cs}}{dt} = -w_s V_{cc} + \frac{i_{rs}}{C_r} \tag{28}$$

$$\frac{dV_{cc}}{dt} = w_s V_{cs} + \frac{i_{rc}}{C_r} \tag{29}$$

$$\frac{dV_{DC}}{dt} = \frac{2\,i_p}{\pi\,C_f} - \frac{V_{DC}}{R_d\,C_f} \tag{30}$$

These equations represent the large signal dynamic model of the DC–DC LLC converter in V2X mode based on the PFM strategy.

When a large signal transient disruption occurs, this model guarantees enough dynamic information for the DC–DC LLC converter. However, this model is used with the PFM strategy that needs wide switching frequencies to cover the entire operating zone. A wide switching frequency causes DC–DC LLC converter saturation, preventing achievement of good control performance and reducing the efficiency. For this reason, this model will be rewritten, in the next section, combined with the PSM strategy that can avoid the PFM limitations and cover the whole operating zone. Furthermore, the proposed model based on PSM strategy allows us to design and apply nonlinear robust controllers in order to ensure robustness of the control against disturbances.

### 2.3.2. PSM

The PSM strategy is proposed to get around the constraint of the PFM strategy in V2X mode in order to improve the DC–DC LLC converter efficiency and control the operating points that the PFM strategy cannot reach in V2X mode [24]. Based on a small signal model with FHA, the PSM strategy in the last section was implemented based on the gain transfer function in V2X mode. Small signal modeling enables creating a transfer function based on an equivalent model, which makes the control simple to implement. It does not, however, account for nonlinear and uncertain effects, such as the DC–DC LLC converter structure, which has an impact on the robustness of the applied control law with regard to system perturbations and results in a constrained operating zone.

Large signal modeling [20,23] has been proposed for the DC–DC LLC converter, based on the PFM strategy, to provide enough dynamic information when large signal transient disturbance occurs. However, this model is used with the PFM strategy that needs wide switching frequencies to cover the entire operating zone, causing LLC converter saturation and reduced efficiency. For this reason, this model has been rewritten and combined with the PSM strategy that can avoid the PFM limitations. Furthermore, the proposed model based on PSM strategy allows us to design and apply nonlinear robust controllers in order to ensure the control robustness against the disturbances.

The equivalent circuit of the DC–DC LLC converter in V2X mode is presented in Figure 5. Based on Figure 5, the square voltage $V_{in}$ represents the input voltage in V2X mode, which depends on the battery voltage $V_b$ and the controlled phase-shift angle $\theta$ in the case of the PSM strategy. Its fundamental component is expressed in Equation (8).

The nonlinear dynamic equations of the DC–DC LLC converter are defined in Equations (11)–(13). Like the large signal model in case of PFM strategy, the resonant tank's AC variables can be divided into sine and cosine components by using the sinusoidal approximation. The following expression (31) presents the resonant circuit's input voltage in V2X mode:

$$V_{in}(t) = V_{if}\sin(w_s t) \tag{31}$$

with $w_s = 2\pi f$ where $f$ is the switching frequency. $V_{if}$ is the RMS value of the fundamental component of the resonant tank's input defined in Equation (8) and is a function of the phase shift $\theta$. The approximation of the series resonant current and its derivative is given in Equations (14) and (15). Likewise, $V_c$ and $\frac{dV_c}{dt}$ can be divided into sine and cosine components, as shown in Equations (16) and (17). The terms $sign(i_r)$ and $|i_r|$ are approximated by their sine and cosine components as shown in Equations (19) and (20).

Using the sinusoidal approximation of each variable as given in Equations (14)–(17), (19), (20) and (31) in the LLC dynamic equations (11)–(13), and splitting the sine and cosine terms, the following equations, (32)–(36), can be obtained:

$$\frac{di_{rs}}{dt} = -w_s i_{rc} - \frac{V_{cs}}{L_r} - \frac{4\,i_{rs}\,V_{DC}}{\pi\,L_r\,i_p} + \frac{n\,V_b}{\sqrt{2}\pi}\sqrt{[10 + 6cos(\theta\pi)]} \tag{32}$$

$$\frac{di_{rc}}{dt} = w_s i_{rs} - \frac{V_{cc}}{L_r} - \frac{4\,i_{rc}\,V_{DC}}{\pi\,L_r\,i_p} \tag{33}$$

$$\frac{dV_{cs}}{dt} = -w_s V_{cc} + \frac{i_{rs}}{C_r} \tag{34}$$

$$\frac{dV_{cc}}{dt} = w_s V_{cs} + \frac{i_{rc}}{C_r} \tag{35}$$

$$\frac{dV_{DC}}{dt} = \frac{2\,i_p}{\pi\,C_f} - \frac{V_{DC}}{R_d\,C_f} \tag{36}$$

These equations present an improved model of the DC–DC LLC converter combined with the PSM strategy in V2X mode. This model describes the LLC dynamics and provides enough information about the resonant tank, the DC bus and the switching network. It provides the necessary information of the LLC dynamics (partially known) to design a nonlinear and robust control law.

A comparison between the small signal modeling and the large signal modeling is conducted in Table 2. The advantages and drawbacks of each modeling approach are highlighted.

**Table 2.** Comparison between small and large signal modeling.

| | Characteristics | Limitations |
|---|---|---|
| Small signal modeling [15,17–19,24,40] | Linearized model with transfer function representation<br>First harmonic approximation with an averaged model<br>Simplified equivalent circuit<br>Simplified control design<br>Ignored effect of harmonics and converter parameters<br>Comfortable with linear controllers | Limited operating zone<br>Lack of realistic model accuracy<br>Lack of dynamic behavior of state variables<br>Not comfortable with nonlinear controller<br>Lack of robustness |
| Large signal modeling [20–23,48,49] | Nonlinear model with higher order<br>Dynamic behavior representation<br>Harmonic approximation<br>Comfortable with nonlinear and robust controllers<br>Closer to the realistic model<br>Cover more operating points | More complex control<br>More complex design<br>Lack of realistic model behavior<br>Still requires approximation methods |

A transfer function based on an averaged or linearized mode can be obtained using small signal modeling, making it simple to implement control. It does not, however, account for nonlinear and unpredictable effects, such as the DC–DC LLC converter structure, which has an impact on the robustness of the applied control law with regard to system disturbances and results in a constrained operating range. From here, the large signal model appears important for obtaining enough information about the DC–DC LLC converter dynamics (partially known) and help with the design of nonlinear and robust control laws.

## 3. Control Strategies Based on the LLC Model

In order to ensure good performance of the DC–DC LLC converter, it is necessary to define a control strategy based on an improved LLC dynamic model that is able to reach all the operating points.

A bidirectional EV charger, as shown in Figure 1, includes an AC–DC topology connected to a DC–DC LLC converter. Regarding the studied charger topology, it is critical to regulate the DC bus voltage when the battery voltage is enforced by the Battery Management System (BMS) with wide variations in voltage (from 240 to 440 V) and power (0–22 KW). The role of the AC–DC converter is to improve the grid quality, reduce the current harmonics and maintain a regulated grid current with a high power factor. The DC bus capacitor is connected to a bidirectional DC–DC LLC resonant converter.

Most research works have studied control solutions for the DC–DC LLC converter related to battery voltage or current control or a dual reaction control with an external voltage control loop and an internal current control loop. In most cases, the battery voltage varies over a reduced range, and the converter operates at a narrow power zone. On the other hand, the studied control strategies for the DC–DC LLC converter in this paper aim to ensure the DC bus voltage control for a bidirectional EV charger application where the battery voltage is constant and imposed by the BMS. It should be noted that the AC–DC converter of the EV charger is used to regulate the grid current and, consequently, the DC bus current at the input of the DC–DC converter. The DC bus voltage is independently regulated by the DC–DC LLC converter to follow the setpoint.

### 3.1. Control Based Small Signal Model

To build a small signal model in power converters, the state space average approach is commonly used. However, because the operating frequency is not constant and the energy is largely conveyed through the fundamental components of the current and voltage signals, it is not appropriate for a frequency-controlled resonant converter.

### 3.1.1. PFM

PFM is generally adopted with the DC–DC LLC resonant converter by varying the switching frequency of the MOSFETs' control signals [44,45]. In [50], a simple equivalent LLC circuit model based on a simplified EDF approach is proposed. A third-order equivalent model is generated. The frequency dynamics can be accurately predicted using the equivalent model. To properly build the feedback control, an analytical study of all the transfer functions has been provided.

The DC bus voltage/battery voltage relationship serves as a representation of the DC–DC LLC transfer function. The gain of the DC–DC LLC converter can be adjusted through a switching frequency control of the H-bridge, which also provides the DC bus voltage control to a specific setpoint (450 V).

For DC–DC LLC converters with large variations in battery voltage (240–430 V) and power (0–11 kW), a Gain Inversion (GI) method for DC bus voltage regulation is described in [24] for V2X mode. In fact, the tuning of the PI controller was found to be complex in order to provide a trade-off of fastness/stability. The advantages of the GI include a faster response as well as trajectory tracking in order to avoid an important current transformer, which may cause a transformer saturation.

The GI allows expressing the switching frequency (feedforward) in the function of the resonant circuit's parameters, DC bus voltage, power request and battery voltage.

The feedforward switching frequency $f_{0d}$, resulting from the GI method, is defined in [24] based on (4).

It should be noted that the switching frequency feasible zone is limited between the minimal authorized switching frequency $f_{min}$ = 60 kHz and the maximal authorized switching frequency $f_{max}$ = 200 kHz.

In Figure 6, $f_{0d}$ is presented according to the battery voltage and power variation.

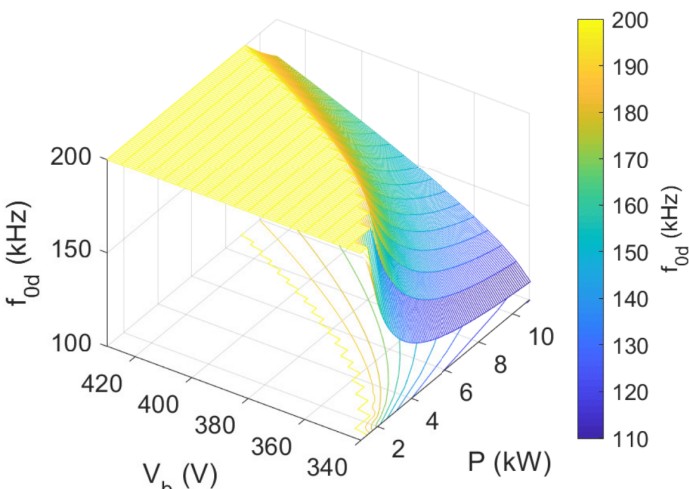

**Figure 6.** Feedforward frequency according to battery voltage and power variation in V2X mode.

Based on Figure 6, there is an important yellow zone that corresponds to a switching frequency of 200 kHz due to the fact that the switching frequency is limited between 60 and 200 kHz. This saturation zone gives low efficiency and low control performance with the PFM strategy in the reversible operation mode of the EV charger. The closed loop PFM control strategy based on the GI in V2X mode [24] is presented in Figure 7.

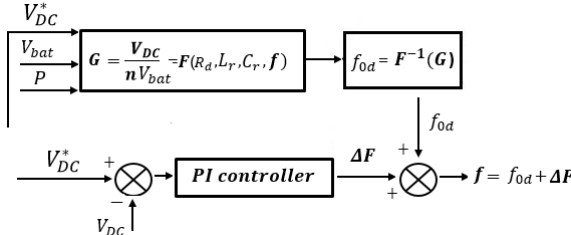

**Figure 7.** Closed loop control block in V2X mode for PFM strategy.

For the entire operating zone, ZVS is needed to provide excellent performance and high efficiency [51]. In order to maintain the ZVS, the switching frequency feasible zone is defined as a range between a minimal and maximum allowed value [32,36]. When it comes to the software implementation and hardware construction of the EV charger, this feasibility constraint presents a cost-cutting challenge. However, by using the PFM strategy, the DC–DC LLC converter has limitations to operate in ZVS for a wide voltage and power variation ranges. There is a significant constraint imposed in front of the control stability over the whole operating zone. These constraints are obviously imposed on the onboard charger. As a result, the DC–DC LLC resonant converters lose their soft-switching characteristics, lowering efficiency and restricting bidirectional power transmission possibilities with the PFM strategy for a wide operating zone.

In order to cover all operating points, the control frequency range, which primarily depends on the resonant tank's components ($L_r$, $C_r$ and $n$), power request ($P$), DC bus voltage ($V_{DC}$) and battery voltage ($V_{bat}$), is insufficient.

It is possible to use a modulation strategy with a fixed switching frequency to control the DC–DC LLC converter in the saturation zone. The PFM approach can be used outside of the saturation zone. The PFM constraint is overcome using PWM and PSM techniques [43,52]. These techniques are suggested to increase the efficiency of the DC–DC LLC converter while extending its feasibility zone at a constant switching frequency [53–55].

### 3.1.2. PWM

In [20,27], sliding mode control is used to insure the battery voltage stability by varying the respective switching frequency and duty cycle. In order to obtain the highest ZVS performance across a wide range of output voltage and load, a two-phase interleaved LLC converter with a simple PWM control and fixed switching frequency is presented in [52]. In [54], a controlled duty cycle–switching frequency modulation approach is studied for a wide voltage range of the DC–DC LLC converter that provides ZVS and output voltage control while improving efficiency under light loading conditions. The best combination of the duty cycle and switching frequency is chosen to provide both ZVS operating and output voltage control.

The voltage gain of the converter is regulated by varying the duty ratio instead of increasing the switching frequency. The closed loop PWM control law based on a small signal model [24] is designed as shown in Figure 8.

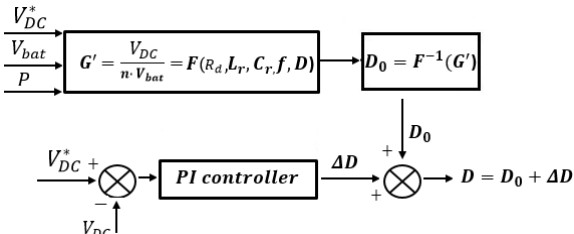

**Figure 8.** Closed loop control block in V2X mode for the PWM strategy.

The feedforward duty cycle $D_0$ is determined using the GI method with a fixed switching frequency $f$ based on PWM gain $G'$ in Equation (7). The static voltage error is subsequently eliminated by adding it to a PI output. The benefit of this gain-inverted control strategy is that the DC–DC LLC converter's duty cycle regulation does not have a broad bandwidth, which avoids a gradual increase in the DC bus voltage during transient conditions.

On the other hand, duty cycle variation needs to be limited in order to comply with FHA and ensure that the DC–DC LLC resonant converter operates properly. FHA is no longer applicable when the duty cycle variation is far from 0.5 [27,56]. In this situation, the ZVS property may also be destroyed. However, with considerable corrective action from the PI controller, a reasonable DC bus regulation error can be achieved. It should be noted that to guarantee ZVS operation with PWM strategy, the duty cycle must change around 0.5.

Thus, when investigating ZVS operation for the whole operation points with PWM strategy, it is necessary to add higher-order harmonic components to the fundamental component for a more precise analysis. The modeling study with more harmonics in the case of PWM strategy leads to a complex model to control.

### 3.1.3. PSM

In [17], a simple PI controller is used to regulate the resonant current flowing in the LLC resonant circuit by varying the phase shift. Ref. [55] examined a dual-phase LLC resonant converter for wide input voltage range applications with phase-shift modulation. In order to control power transfer and keep the output voltage constant while the input voltage fluctuates, phase-shift control was used. For PV applications, Ref. [57] developed a dual-input single resonant tank LLC converter. The authors implemented independent maximum power point tracking (MPPT) for each panel using a phase-shift PWM control. A fixed frequency phase-shift-modulated LLC resonant converter was proposed in [34]. A phase shift was added between the square waves, with frequencies that are always tuned to the resonant frequency, to control the output voltage.

The closed loop control law based on a small signal model [24] is designed in Figure 9.

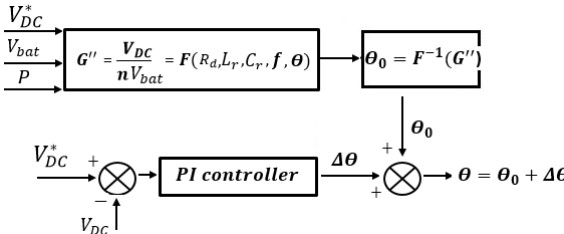

**Figure 9.** Closed loop control block in V2X mode for PSM strategy.

Based on the PSM gain $G''$ in Equation (10), the feedforward phase-shift angle $\theta_0$ is defined by the GI. The phase shift $\theta_0$ is added to the output of a PI controller in order to ensure the DC bus voltage control around a defined request $V_{DC}^*$.

Without using a soft-start strategy, the PSM strategy based on GI enables a gradual increase in the DC bus voltage toward its setpoint with an acceptable overshoot percent tolerance.

### 3.1.4. Comparative Study

The parameter settings are shown in Table 3.

**Table 3.** Table of LLC parameters and control gains.

| | | | |
|---|---|---|---|
| $C_f(\mu F)$ | 75 | $K_p(PFM)$ | 40 |
| $C_r(\eta F)$ | 80 | $K_i(PFM)$ | 1 |
| $L_r(\mu H)$ | 30 | $K_p(PWM)$ | 0.00005 |
| $L_m(\mu H)$ | 120 | $K_i(PWM)$ | 0.00001 |
| $n$ | 1.6 | $K_p(PSM)$ | 0.005 |
| $\epsilon(V)$ | 5 | $K_i(PSM)$ | 0.0001 |

In this section, the different LLC modulation strategies PFM, PWM and PSM are compared in V2X mode in terms of control system performance (overshoot percent, rising time and tracking error) and DC–DC LLC converter efficiency as presented in [24], based on a small signal model.

Figures 10 and 11 show an efficiency comparison between PFM, PWM and PSM strategies according to the power variation for some battery voltages.

Figures 10 and 11 illustrate the high efficiency degradation with the PFM approach at low powers (high switching losses caused by high switching frequency demand). It is obvious that the PWM approach was unable to increase the DC–DC LLC converter efficiency at low power loads because the duty cycle varied outside of the ZVS condition associated to FHA. In comparison to PFM and PWM strategies, the PSM strategy, with fixed switching frequency and fixed duty cycle, has an enhanced efficiency of approximately 87% (With $V_{bat}$ = 350 V in Figure 10) and 84% (With $V_{bat}$ = 420 V in Figure 11) for P = 2000 W.

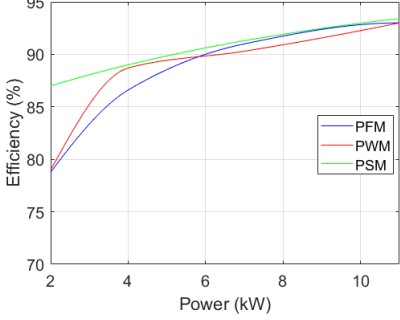

**Figure 10.** Efficiency comparison for $V_{bat}$ = 350 V.

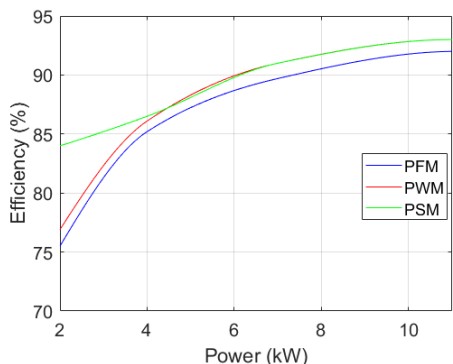

**Figure 11.** Efficiency comparison for $V_{bat}$ = 420 V.

PFM strategy exhibits a high efficiency at high powers, with $V_{bat}$ = 350 V in Figure 10, which is the case of operating points presented in the non-saturated control frequency zone (outside the yellow zone in Figure 6), while PWM and PSM strategies are more efficient than the PFM strategy at $V_{bat}$ = 420 V at high powers in Figure 11, which is the case of the saturation zone.

It is verified that the PWM strategy has good control performance, but efficiency at low power loads does not increase because the FHA model conditions are not met. In contrast to the PWM strategy, the PSM strategy, with a fixed duty cycle of 0.5, does not have an FHA constraint and can avoid the drawbacks of the PFM strategy with good control performance and a high converter efficiency for the entire operating zone.

To summarize, the existing control laws in the literature based on a small signal model, combined with the different modulation strategies, are applied with the popular PI controller.

On the other hand, the linear controllers, such as the PI and GI, require the knowledge of the DC–DC LLC model and react sensibly to unexpected disturbances. Therefore, nonlinear control laws have been proposed in order to ensure the control robustness against the disturbances with unknown boundaries.

### 3.2. Control Based Large Signal Model

In the literature, an improved model of the DC–DC LLC converter based on the PFM strategy has been designed. However, the application of different control laws for such a model provides switching frequency saturation for a wide operating zone in V2X mode. This PFM limitation is avoided by applying the PSM strategy based on a small signal model. The small signal model is not sufficient for describing the DC–DC LLC converter dynamics. Linear control laws, such as GI [24] and PI [33] controllers, are popularly adopted with the PSM strategy. These controllers require the knowledge of the DC–DC LLC model. However, in the case of any unexpected disturbance, these control laws cause an oscillatory behavior and an important tracking error, which results in a current oscillation in the transformer as well as overheating. This provides weak control performance where the DC–DC converter is implemented with the AC–DC converter in the EV charger. From here, the importance of nonlinear and robust controllers, which can be implemented without the knowledge of the unexpected disturbances (current and voltage disturbance, parameter variation), becomes apparent.

A general comparison between linear and nonlinear control laws, which have been implemented in the literature for the DC–DC LLC converter in V2X mode, is conducted in Table 4.

**Table 4.** Comparison between linear and nonlinear control laws.

| | Characteristics | Limitations |
|---|---|---|
| Linear controllers (feedback [23], control based observer [49], PI [33,37], GI [24]) | Based on small signal model<br>Simple control design<br>Less cost in implementation<br>Non-complicated gain tuning | Less robust versus disturbances<br>Lack of precision over the whole operating zone<br>Low control performance for many operating points |
| Nonlinear controllers (Sliding mode [20,21], MFC [46], ASTC [47], MPC [58]) | Based on large signal model<br>More robust versus disturbances<br>Better control performance<br>Improved efficiency | More complex control design<br>Higher cost in implementation<br>Complex gain tuning |

### 3.2.1. Model Predictive Control (MPC)

In order to decrease power losses, the series resonant converter is thought to be controlled by MPC, as this control method optimizes the choice of the converter switching state. An overview of MPC applied to power converters is in [59].

Because MPC strategies depend on the model of the system (parameters) and the measurements, the behavior of the control approach-based MPC can be impacted by inaccuracies in the model parameters and/or noise in the measurements. An integration of all constraints presented in the system in the control algorithm can improve the MPC robustness. Additionally, these constraints are challenging to apply using traditional linear control approaches, which makes the controller more complex. Because of these factors, MPC is investigated as a useful control strategy to give resonant power converters better control and adaptability [60].

The MPC was applied to the DC–DC LLC converter in [58] avoiding the use of a modulator. The output voltage is controlled using a predictive control strategy in order to improve the converter efficiency under light load conditions.

### 3.2.2. Model Free Control (MFC)

To "robustify" a priori any "unknown" or "poorly known" dynamical systems for which uncertainties of the model parameter(s) and unexpected disturbances are considered, Fliess & Join [61] introduced the Model Free Control (MFC) concept. Within its working range, a finite-dimensional ordinary differential equation (37), which is not always linear, controls the system's input–output behavior:

$$E(y, \dot{y}, \ldots, y^{(a)}, u, \dot{u}, \ldots, u^{(c)}) = 0 \qquad (37)$$

where $a$ is the system order ($c \leq a$), $u$ is the control input and $y$ is the system output. The following ultra-local model can replace the complicated nonlinear system over a very brief time interval (38) [61]:

$$y^{(\nu)}(t) = F + \alpha u(t) \qquad (38)$$

where $y^{(\nu)}$ is the derivative of order $\nu$ of $y$, and $\nu$ is the relative degree of the system. $F$ contains all the structural information that depends on the other variables of the system (including disturbances and nonlinearity). $\alpha$ is a constant parameter (with non-physical sense) selected by the practitioner selects such that $u$ and $y^{(\nu)}$ have the same magnitude [62]. There is no need to differentiate between the poorly known model parameters and the disturbances because the parameter $F$ is continuously updated and encompasses both. The control can be updated in real time by estimating the numerical value of $F$ [62].

The MFC is given as in Equation (39) [61]:

$$u(t) = \frac{1}{\alpha} \left[ y^{*(\nu)}(t) - \hat{F} + R(e(t)) \right] \qquad (39)$$

where $y^*$ is the reference and $e(t)$ is the tracking error. $R$ is a classical PI regulator, i.e., $R(e(t)) = K_p e(t) + K_I \int e(t)dt$.

The DC–DC LLC converter is a nonlinear system resulting from many parameters: nonlinear dynamic relation between AC (input of the LLC resonant circuit) and DC (DC bus or battery voltage) variables due to MOSFET switching operating, inductor currents, series capacitor voltage and square voltage across the transformer containing switching frequency harmonics.

For the DC–DC LLC converter, $y$ is the measured DC bus voltage, $y^*$ is the DC bus voltage request and $u$ is the controlled phase shift $\theta$. Then, the complex nonlinear LLC converter system can be written, by taking $v = 1$ in Equation (38), as in Equation (40):

$$\dot{V}_{DC} = F + \alpha\theta(t) \tag{40}$$

$\dot{V}_{DC}$ is the first order derivative of $V_{DC}$. As studied in [46], the controlled phase shift can be derived using the MFC strategy as in Equation (41):

$$\theta(t) = \underbrace{\frac{-\hat{F}}{\alpha}}_{\text{Robustness insured}} + \underbrace{\frac{1}{\alpha}\left(\dot{V}_{DC}^* + K_p e(t) + K_I \int e(t)dt\right)}_{\text{Closed loop tracking}} \tag{41}$$

The MFC law is divided into two parts. The first part aims to cancel the effect of the nonlinearity and the disturbances, and the second aims to track the setpoint in the closed loop.

The control block with the MFC strategy, discussed in [46], is presented in Figure 12.

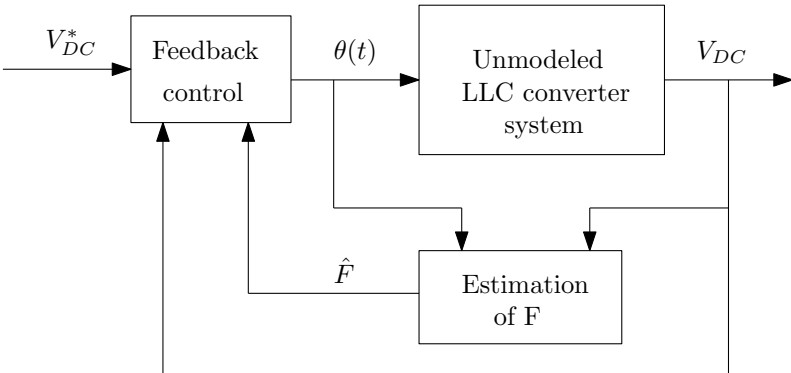

**Figure 12.** MFC strategy block.

As a result, the controlled phase shift derived from the MFC strategy can only be acquired based on the DC bus voltage measurement and the DC bus voltage request when the PSM strategy is applied to the DC–DC LLC converter. The proposed control strategy, which is based on the MFC, allows for the tracking of the DC bus voltage request and offers robustness against system disturbances.

### 3.2.3. Adaptive Super Twisting Control (ASTC)

The sliding mode control is well known in the family of robust control laws with respect to system disturbances. The ASTC principle has been investigated in [63,64]. The ASTC has been developed for situations when the bounds of disturbances and uncertainty are unknown. The suggested control rule combines Super Twisting Control (STC) with dynamically adaptive control gains. The fundamental aspect of the adaptation method is that the control gains are not overestimated. The ASTC major characteristic is that it does not only ensure DC bus voltage reference tracking but also provides robustness against system disturbances. The PSM method works by establishing a phase shift between the MOSFET control signals with a duty cycle of 0.5 and a switching frequency of 200 kHz for

all MOSFETs. The new challenge of [47] was to obtain the controlled phase shift using the ASTC law in order to improve the control performance.

An improved model of the DC–DC LLC converter based on the PSM strategy is studied in the last section. The aim of this model is to give enough information of the LLC dynamics (partially known). Moreover, it will help to design a nonlinear robust control. Nonlinear control laws implemented in the literature in order to ensure the control robustness against the disturbances with unkown boundaries are presented in this section.

In Equations (32) and (33), $\frac{di_{rs}}{dt}$ and $\frac{di_{rc}}{dt}$ are respectively defined. Since the controlled phase shift appears in Equation (32), it can be observed that the control input indirectly appears in the derivative of $i_p$. As a result of Equations (32), (33) and (36), $\ddot{V}_{DC}$ can be written as in (42):

$$\ddot{V}_{DC} = A(x,t) + B(x,t)u \tag{42}$$

where $x = \begin{bmatrix} i_{rs} & i_{rc} & V_{cs} & V_{cc} & V_{DC} \end{bmatrix}$ is the vector of the state variables defined in the five Equations (32)–(36) of the DC–DC LLC converter model. $A(x,t)$ and $B(x,t)$ are defined in [47]. $A(x,t)$ and $B(x,t)$ depend on the internal dynamics, which are considered bounded, and they can be presented as bounded disturbances/uncertainties with unknown boundaries.

The sliding variable can then be chosen, based on (42), as follows (43):

$$\sigma = e + k\dot{e} \tag{43}$$

where $e = V_{DC} - V_{DC}^*$ is the tracking error, $V_{DC}^*$ is a constant DC bus voltage request, and $k$ is a constant that aims to adjust the convergence speed of the DC bus voltage tracking error $e$ when $\sigma$ is equal to zero (in the sliding surface). The ASTC law is defined in Equations (44)–(46) as detailed in [63]:

$$u = -\alpha |\sigma|^{0.5} sign(\sigma) - \int \beta sign(\sigma) dt \tag{44}$$

$$\dot{\alpha} = \begin{cases} w_1 sign(|\sigma| - \mu) & if\ \alpha > \alpha_m \\ \eta & if\ \alpha \leq \alpha_m \end{cases} \tag{45}$$

$$\beta = 2\epsilon\alpha \tag{46}$$

The controlled phase shift for the PSM approach can be derived from the ASTC as presented in [47]. The control block of the PSM approach combined with the ASTC [47] is presented in Figure 13.

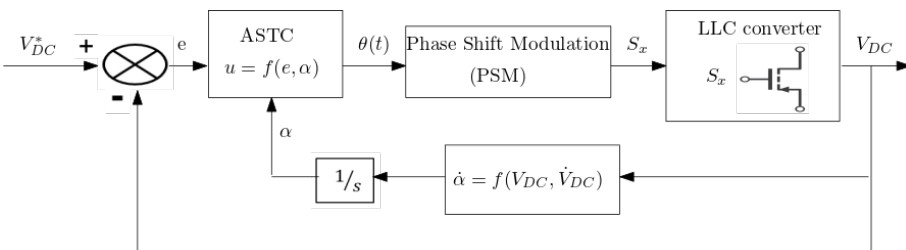

**Figure 13.** PSM combined with ASTC strategy block.

As a result, based on the DC bus voltage request and the DC bus voltage measurement, the controlled phases shift is derived from the ASTC law. The phase-shift angle is used to generate the MOSFETs' control signals ($S_x$, where $x$ goes from 1 to 8) for the DC–DC LLC converter in order to apply the PSM approach in V2X mode for the entire operating zone.

3.2.4. Comparative Study

In [47], a comparative study of control strategies based on PI [17,33,34], STC, GI method [24] and ASTC [47], which are all applied to the PSM approach, has been conducted.

The results reveal that the ASTC has a benefit in terms of ensuring the control robustness against disturbances as well as improving control performances and LLC converter efficiency.

The new challenge of this section is to compare linear and nonlinear control strategies applied to the DC–DC LLC converter under different scenarios. The following results are obtained for $P$ = 2000 W and $V_b$ = 420 V. The DC bus voltage request is 450 V, with a maximum authorized voltage tracking error of 10 V.

Comparison with Respect to a DC Bus Current Disturbance

A disturbance with a magnitude of 2 A and a frequency of 100 Hz is added to the DC current, flowing through the equivalent resistor $R_d$ (Figure 5), at 0.04 s. Figure 14 shows a comparison of the DC bus voltage responses for the various controllers.

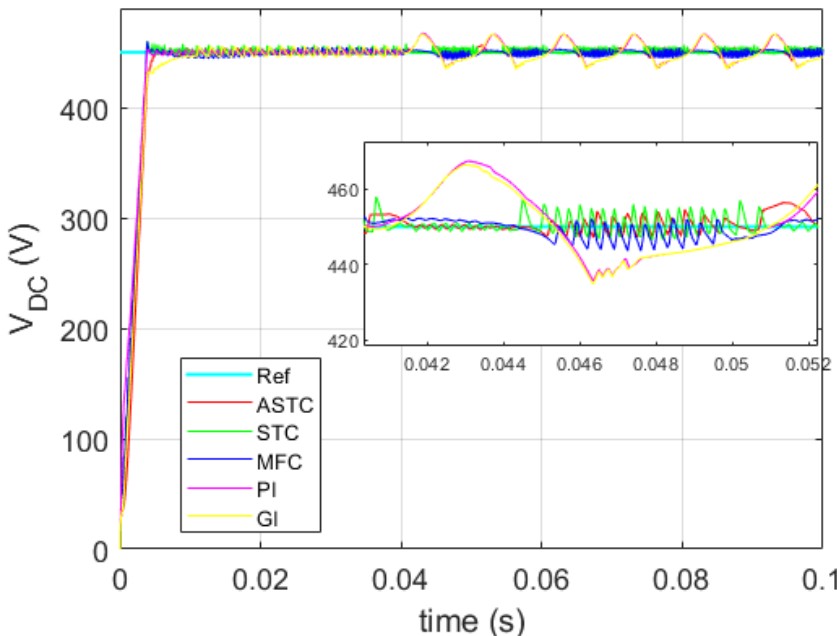

**Figure 14.** DC bus voltage comparison with the current disturbance.

In comparison to the GI and PI controllers, the proposed ASTC, such as STC and MFC, is clearly robust with respect to the presented disturbance. The GI and PI are linear controllers that immediately depend on the LLC model, which makes them less robust.

On the other hand, the conduction losses are mainly computed based on the RMS value of the DC current. In order to see the effect of different controllers on the conduction losses, Figure 15 shows the series inductor current $i_r$ for the different strategies. According to this figure, PI and GI exhibit a significant DC bus voltage tracking error even though they have the lowest maximal RMS value of the resonant current $i_r$. On the other hand, it is evident that the ASTC, due to chattering reduction and adaptive ASTC gain in the presence of the current disturbance, delivers lower maximal RMS current (peak value of 22.2 A) than both MFC (peak value of 26.4 A) and STC (highest value of 26.5 A). Due to this ASTC benefit, the DC–DC LLC converter's conduction losses can be reduced.

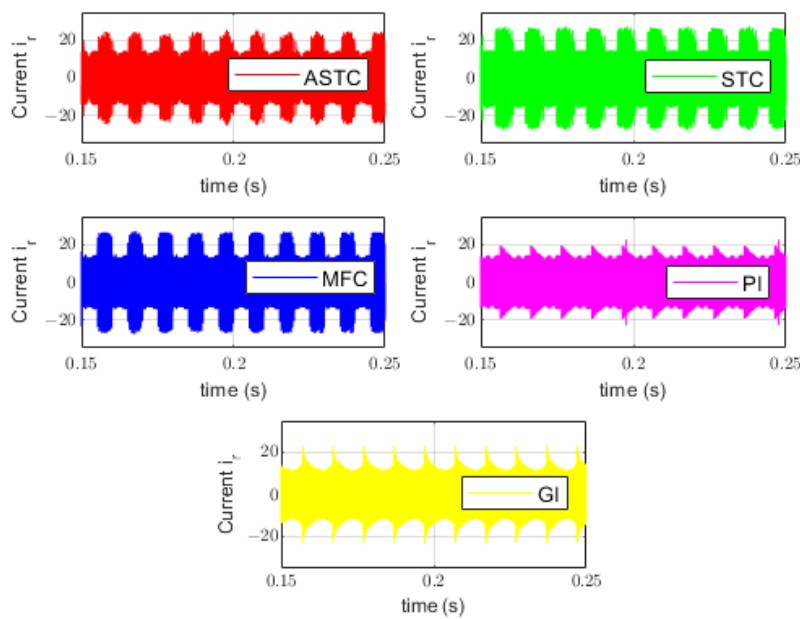

**Figure 15.** Resonant current with the current disturbance.

Figure 16 displays a comparison table that contrasts the DC bus voltage responses of the various controllers in terms of control performance and efficiency.

| $V_b$=420 V, P=2000 W | GI | PI | MFC | STC | ASTC |
|---|---|---|---|---|---|
| Maximal DC bus voltage tracking error (V) | 18 | 17 | 5 | 6 | 5 |
| Settling time (s) | 0.01 | 0.004 | 0.004 | 0.0039 | 0.0045 |
| Rising time (s) | 0.0038 | 0.0032 | 0.0033 | 0.00285 | 0.0029 |
| Overshoot percent (%) | 0 | 0.6 | 0 | 2.2 | 1.3 |
| Efficiency (%) | 75 | 74 | 77 | 69.5 | 79 |
| Control's energy consumption index J | 0.0107 | 0.0142 | 0.0187 | 0.0253 | 0.0173 |

**Figure 16.** Comparison table with the current disturbance.

The *J* index can be thought of as the control system's energy consumption index, i.e., related to the switching losses of the DC–DC LLC converter. It is denoted by the formula $J = \sum_{k=1}^{m} \theta^2(k)$ (*m* stands for the final time). It has the energy of the regulated phase-shift angle.

The controlled phase shift with STC produces the highest frequency oscillation in the presence of a disturbance due to the chattering phenomenon of the sliding mode. Moreover, the MFC presents higher frequency oscillation than the ASTC. In fact, the adaptive control gain *α* of the ASTC adapts in order to reject the effect of the disturbance. The chattering in ASTC is reduced, and the controlled phase shift has a significantly lower frequency oscillation than that in STC and MFC, which have constant gains, resulting in reduction of the switching losses. It should be noted that both MFC and STC have constant control gains, but the chattering caused by the sign function of the STC leads to it having higher control frequency oscillation than the MFC.



Comparison according to the Trajectory Tracking

Here, the disturbance with an amplitude of 40 V and a frequency of 100 Hz is injected to the voltage request at 0.03 s. As illustrated in Figure 17, the ASTC, such as STC and MFC, can guarantee the DC bus voltage request tracking. Because they are directly impacted by parameter change, the GI and PI approaches have an important tracking error, which makes them less robust.

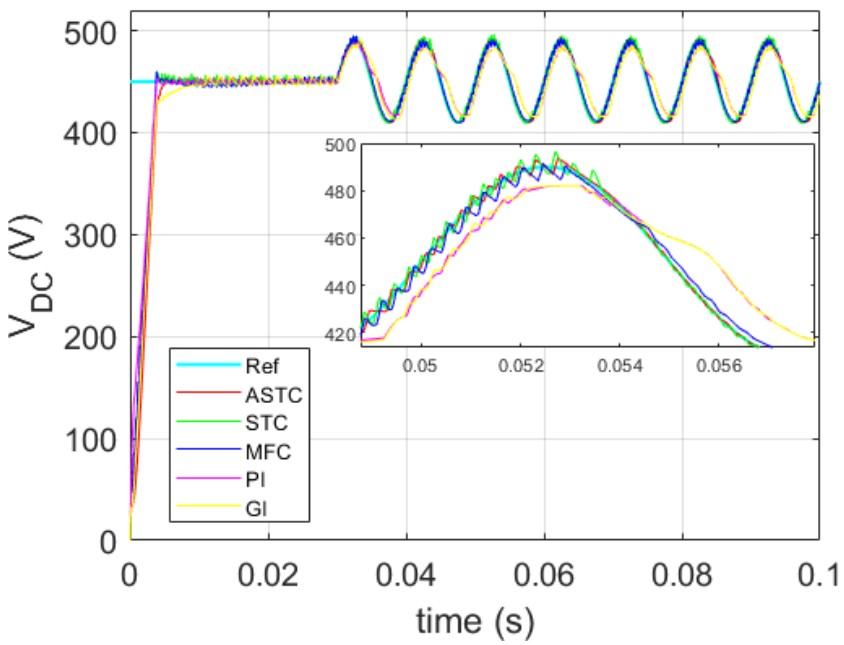

**Figure 17.** Voltage response comparison with varying request to be tracked.

As can be shown in Figure 18, the ASTC offers lower RMS series resonant current (peak value reaches 26 A) than MFC (peak value of 28 A) and STC (peak value of 31.5 A).

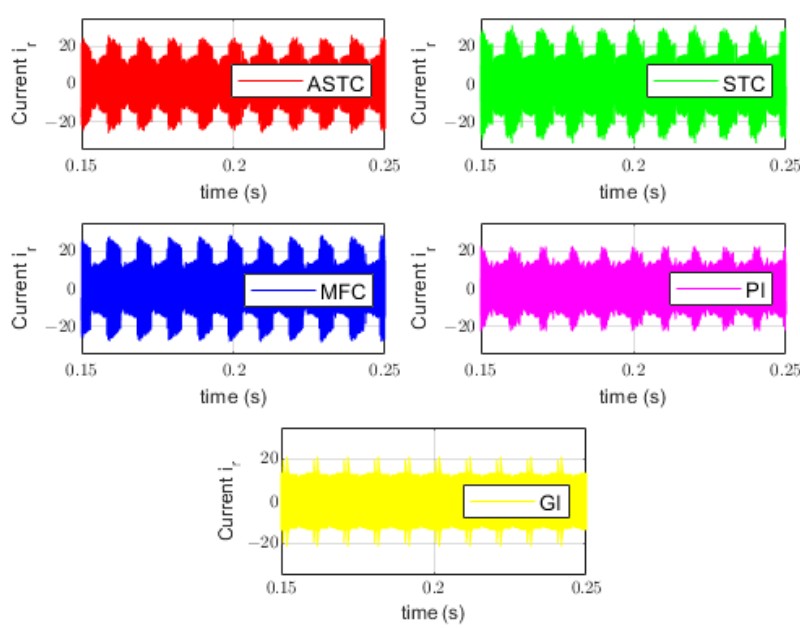

**Figure 18.** Resonant current with varying request to be tracked.

Figure 19 presents a comparison table comparing the various controllers with the varying request to be tracked.

| $V_b$=420 V, P=2000 W | GI | PI | MFC | STC | ASTC |
|---|---|---|---|---|---|
| Maximal DC bus voltage tracking error (V) | 20 | 20 | 5 | 6 | 6 |
| Settling time (s) | 0.01 | 0.0034 | 0.0044 | 0.0037 | 0.0047 |
| Rising time (s) | 0.0036 | 0.0036 | 0.0034 | 0.0034 | 0.0037 |
| Overshoot percent (%) | 0 | 0.6 | 2.2 | 1.1 | 0 |
| Efficiency (%) | 68.1 | 68.5 | 75 | 59 | 79 |
| Control's energy consumption index J | 0.0149 | 0.0145 | 0.0187 | 0.0214 | 0.0181 |

**Figure 19.** Comparison table with varying request to be tracked.

Table 5 displays the advantages and drawbacks of different linear and nonlinear control laws applied to the DC–DC LLC converter in combination with the PSM strategy.

**Table 5.** Comparative study of controllers combined with the PSM strategy.

| | Characteristics | Limitations |
|---|---|---|
| PI [17,33,34] | Simple control design<br>Simple gain tuning<br>Low implementation cost | Requires model knowledge<br>Less robust versus disturbances |
| GI [24] | Avoid wide control bandwidth<br>Variation around equilibrium point<br>Simple control design<br>Simple gain tuning<br>Low implementation cost | Requires model knowledge<br>Less robust versus disturbances<br>Requires equilibrium point |
| MPC [58] | More robust than linear control<br>Avoid modulator system<br>Optimization of the switching state | More complex control design<br>Depends on the system model<br>Needs integration of all constraints |
| MFC [46] | Simplified model design<br>Based on estimation of a black box model<br>Simple gain tuning<br>More robust control than linear control | Requires the system relative degree<br>Requires the output derivative<br>Sensible to noises<br>High implementation cost |
| STC [47] | Based on sliding mode control<br>More robust control than linear control<br>Non-complicated gain tuning | Chattering phenomenon<br>Requires model knowledge<br>Higher switching losses<br>Higher control gains |
| ASTC [47] | More robust control than linear control<br>Adaptive control gains<br>Reduced chattering phenomenon<br>Avoid gain overestimation<br>Lower switching losses | More complex gain tuning<br>High implementation cost<br>Requires a system relative degree equal to 1 |

## 4. Conclusions

As part of the energy transition, the technology of bidirectional chargers for EVs is gaining importance. The EV onboard charger is of two parts: AC–DC converters and DC–DC converters. Work has focused on the bidirectional DC–DC LLC resonant converter in order to improve charger efficiency across both battery power and voltage ranges in V2X mode. Several control strategies, proposed in the literature for the LLC resonant converter, have been studied. Different modulation strategies, such as PFM, PWM and PSM, applied to the DC–DC LLC converter in V2X mode, have been discussed. In order to implement control strategies, two modeling approaches for LLC converter, large and small signal

modeling, have been presented. The principle of each modeling process was defined. The advantages and drawbacks were highlighted.

The small signal model concept applied to the DC–DC LLC converter in V2X mode is used in order to apply linear controllers, based on the PI, combined with the three modulation strategies. Several modulation strategies of the DC–DC LLC resonant converter, in V2X mode, have been presented according to an averaged small signal model based on the FHA. The drawbacks of the PFM strategy result in the push to implement fixed switching frequency strategies, such as PWM and PSM. The benefits and drawbacks of each strategy have been studied. The results show the advantages of the PSM strategy in improving the DC–DC LLC converter control performance and efficiency for the entire operating zone in V2X mode. On the other hand, the large signal model combined with PFM strategy was firstly studied. However, due to limitations of the PFM strategy, an improved model combined with the PSM strategy was then highlighted in order to apply nonlinear controllers. The main idea of this model is to propose robust control strategies in order to ensure robustness against different disturbances and uncertainties in the DC–DC LLC converter system.

Perspectives for future work include the study of advanced modulation strategies for different AC–DC converter topologies used besides the DC–DC LLC converter for bidirectional EV chargers in order to conduct a comparative study in terms of control performance, efficiency, total harmonic distortion and design complexity.

**Author Contributions:** Conceptualization, H.A.A.; methodology, H.A.A., M.G. and M.T.; software, H.A.A.; writing—original draft preparation, H.A.A.; writing—review and editing, M.G. and M.A.H.; visualization, M.G., M.A.H. and M.T.; data curation, H.A.A.; supervision, M.G., M.A.H. and M.T.; Validation, M.G. and M.T. All authors have read and agreed to the published version of the manuscript.

**Funding:** This work was supported by the project chair between Renault Group and Centrale Nantes on electric vehicle bidirectional charger control.

**Data Availability Statement:** The study did not report any data.

**Conflicts of Interest:** The authors declare no conflict of interest.

### Nomenclature

| | |
|---|---|
| $EV$ | Electric vehicle |
| $G2V$ | Grid to Vehicle |
| $V2X$ | Vehicle to Everything |
| $L_r$ | Series inductor of LLC converter |
| $C_r$ | Series capacitor of LLC converter |
| $L_m$ | Parallel inductor of LLC converter |
| $ZVS$ | Zero-Voltage Switching |
| $n$ | Transformer ratio |
| $V_{DC}$ | DC bus voltage |
| $V_{bat}$ | Battery voltage |
| $P$ | Converter power |
| $FHA$ | First Harmonic Approximation |
| $PFM$ | Pulse Frequency Modulation |
| $PWM$ | Pulse Width Modulation |
| $PSM$ | Phase-Shift Modulation |
| $PI$ | Proportional Integral |
| $GI$ | Gain Inversion |
| $MFC$ | Model Free Control |
| $STC$ | Super Twisting Control |
| $ASTC$ | Adaptive Super Twisting Control |

| $f_{0c}$ | Feedforward switching frequency in G2V |
| $f_{0d}$ | Feedforward switching frequency in V2X |
| $f_{min}$ | Minimum authorized switching frequency |
| $f_{max}$ | Maximum authorized switching frequency |

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
