# Peer review of "Review on Modeling and Control Strategies of DC–DC LLC Converters for Bidirectional Electric Vehicle Charger Applications"

_energies, doi:10.3390/en16093946_

Round 1
Reviewer 1 Report
The paper presents a literature review of different modulation strategies (PFM, PWM, PSM) and modelling approaches for DC-DC LLC converter in V2X mode. The review could be useful addition to literature, however I have the following comments:
- My main comment is that the conclusion/summary can be improved, esp. given it is a review paper. It is not clear how the model or expressions corresponding to different modulation strategies is useful or help in providing insights about the pros and cons mentioned.
For instance, at the end of Section 2.1, the authors mention: "To summarize, the main characteristics and limitations of different modulation strategies, applied to the DC-DC LLC converter for an EV charger application in V2X mode, are shown in Table 1." It is not clear how the properties/limitation mentioned in Table 1 are related to the model (e.g. gain) derived earlier in the section? (same is the case with many of the other tables)
The authors are requested to elaborate how the mentioned model expressions are useful for the readers and relate to the properties of LLC converters mentioned. If possible, this could be done via quantitative example or summarize by adding a discussion on this aspect e.g. elaborate on how the models can be useful in ensuring robustness of a design.
- Please re-review the paper organization. In a few places the section content is not in sync with the title. For example, Section 1.2 titled "system presentation" is misleading. Only DC-DC converter is discussed here. The system in Figure 1 has other components such as AC-DC converter, filter, too.
- Many of the figures are a little blurry.
Reviewer 2 Report
The article gives an overview of modelling and control for bidirectional DCDC power converters with galvanic isolation. The topic is certainly interesting and topical.
Below are some technical considerations and tips on how to improve the manuscript.7
I strongly recommend reorganising the introductory section to make it more user-friendly for the reader.
I recommend the following subsections:
- Motivation
Here the authors should further motivate the importance of the MBD approach for the design and control of power electronic systems, with particular emphasis on automotive applications.
In this regard, I recommend reinforcing the motivation with the following works on related power electronics systems.
https://www.mdpi.com/2079-9292/10/23/2954
https://www.mdpi.com/1996-1073/15/23/8990
This strengthens the justification for using MBD approaches for advanced systems of modern research interest.
- Related works/state of the art overview
in the current version of the manuscript, there is actually no section devoted to an overview of the state of the art specific to the modelling and control of power electronics systems and in particular DCDC converters with galvanic isolation.
Here, it would be advisable for the authors to increase the analysis of the most recent articles, with particular emphasis on the chosen LLC architecture.
Please include citations such as the following (looking for more, of course):
https://ieeexplore.ieee.org/abstract/document/9446870
https://www.mdpi.com/2079-9292/11/1/112
- Contributions
Reorganise the section in the current version of the manuscript, making it more explicit what the claims of the work presented are and how it augments the state of the art.
The "system presentation" section should be moved to the modelling part as an introduction to the choice of this particular circuit configuration. In fact, I recommend inserting a few comments comparing possible circuit configurations of DCDC converters with galvanic isolation, such as CLLC and DAB, and why, according to the authors, the choice made has advantages (and possible disadvantages as there are objectively).
Introduce the description of triangular and trapezoidal modulations, commenting on advantages and disadvantages between the various modulation techniques. In addition, I recommend collecting modulation techniques in one subsection "modulation system" and summarising the comparison in a dedicated table.
Figures 7, 8 and 9 show the control loops schematically.
Please comment and justify adequately the lack of current control, which is usually used in the context of a dual reaction control with an external voltage control loop and an internal current control loop and thus a higher bandwidth.
The model was used to deduce the values of the proportional and integrative gains of the control blocks. Was a variational analysis done in simulation?
In the set of controllers that exploit both the linearised and non-linear models, please also mention predictive control, which is very often applied in this type of application.
I hope you find these comments useful.
I look forward to the new version of the manuscript.
Reviewer 3 Report
The review is interesting but, this reviewer thinks that the manuscript will be improved by including more references and comparing the schemes under review in detail including power losses, switching, and conduction losses.
What happened whit the complexity of the schemes?
What kind of platforms are used to implement the control schemes?
Is it possible to include simulation results under different scenarios?
Round 2
Reviewer 2 Report
The new version of the manuscript its integrates all the comments i provided to the authors.
I think it is fully compliant with MDPI publication standards.
Reviewer 3 Report
The manuscript presents an improved version, and the recommendations suggested have been attended to.